# Generative Posterior Networks for Approximately Bayesian Epistemic Uncertainty Estimation

## Abstract

Ensembles of neural networks are often used to estimate epistemic uncertainty in high-dimensional problems because of their scalability and ease of use. These methods, however, are expensive to sample from as each sample requires a new neural network to be trained from scratch. We propose a new method, Generative Posterior Networks (GPNs), a generative model that, given a prior distribution over functions, approximates the posterior distribution directly by regularizing the network towards samples from the prior. This allows our method to quickly sample from the posterior and construct confidence bounds. We prove theoretically that our method indeed approximates the Bayesian posterior and show empirically that it improves epistemic uncertainty estimation over competing methods.

## 1 Introduction

Deep learning systems are becoming ubiquitous in many industries, specifically in applications with large datasets and where prediction mistakes are inexpensive. However, for many other industries, such as autonomous manufacturing, where datasets are smaller and mistakes can be catastrophic, the adoption of deep learning techniques has been slow. One of the critical challenges to applying deep learning models in these safety-critical domains is the lack of uncertainty estimation. If a model makes a prediction in a safety critical environment, we would like that model to be highly confident that prediction is correct. Uncertainty prediction can also provide significant benefit in data-poor domains by guiding exploration towards areas of uncertainty. Most deep learning models, however, are not able to estimate epistemic uncertainty – uncertainty deriving from lack of data samples. In this work, we introduce a new method for modeling epistemic uncertainty in neural networks.

We formulate this problem as a Bayesian Inference problem: given a prior distribution over functions and a set of training data, we want to model the posterior function distribution. There are many existing approaches to constructing such a posterior, such as Gaussian Processes (GPs) or Variational Inference (VI) methods, but these methods can require significant engineering to work in practice and can be expensive in high-dimensional settings. For this reason, many practitioners turn to using

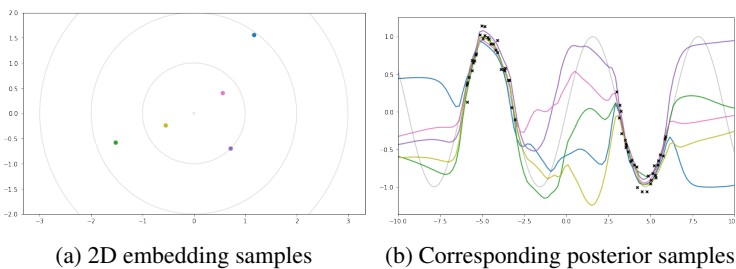

(a) 2D embedding samples    (b) Corresponding posterior samples

Figure 1: Samples from a Generative Posterior Network using a 2D embedding trained on a simple sine-function. On the left are samples from the embedding with corresponding posterior samples on the right. Black x̀s represent observed data points.

ensembles of neural networks to approximate epistemic uncertainty for their ease of use in high-dimensional settings. Pearce et al. (2020) illustrated that, given the correct regularization, these ensemble members approximate samples from the posterior distribution of functions. However, using this technique, every new sample of the posterior requires training a new neural network from scratch, which can be prohibitively expensive for large problems.

To address this challenge, we introduce Generative Posterior Networks (GPNs), a generative neural network model that directly approximates the posterior distribution by regularizing the output of the network towards samples from the prior distribution. By learning a low-dimensional latent representation of the posterior, our method can quickly sample from the posterior and construct confidence intervals. We prove that our method approximates the Bayesian posterior over functions and show empirically that our method can improves uncertainty estimation over competing methods.

## 2 RELATED WORK

There are two categories of uncertainty in the context of probabilistic modeling: aleatoric uncertainty and epistemic uncertainty. Aleatoric uncertainty refers to uncertainty inherent in a system, even when the true parameters of the system are known, while epistemic uncertainty refers to modeling uncertainty that can be reduced through collecting more data (Der Kiureghian & Ditlevsen, 2009; Gal, 2016). Deep learning models have shown impressive performance in quantifying aleatoric uncertainty when given enough data. On the other hand, despite many methods seeking to estimate epistemic uncertainty due to its well-motivated applications in Bayesian Optimization (Snoek et al., 2012; Springenberg et al., 2016), Reinforcement Learning (Curi* et al., 2020), and adversarial robustness (Stutz et al., 2020), quantifying it remains a challenging problem for deep learning models.

Gaussian Processes (GPs) remain one of the most effective methods for epistemic uncertainty estimation in low-dimensional, low-data settings. While scalability can be improved with interpolation methods (Hensman et al., 2013; Titsias, 2009; Wilson & Nickisch, 2015) and kernel learning methods (Wilson et al., 2016), GPs tend to perform worse than neural network-based methods on high-dimensional problems.

Another way to formulate epistemic uncertainty estimation is as a Bayesian Inference problem. Specifically, given a prior distribution over functions and some labeled data, the goal is to construct a posterior distribution in the Bayesian sense. Because this posterior distribution is usually intractable to compute exactly, approximate sampling methods are used instead. Markov-Chain Monte Carlo (MCMC) (Tanner & Wong, 1987) and Variational Inference (VI) (Blei et al., 2017) are some of the most common methods for posterior sampling. These methods, however, tend to struggle in high dimensional problems. The computational complexity of MCMC scales poorly with the dimension of the data and VI methods tend to require restrictive function families with strong independence assumptions. Bayesian Neural Networks (BNNs) (Blundell et al., 2015) are another way to perform Bayesian Inference a neural networks by modeling each parameter in the network with an independent Gaussian distribution. While BNNs perform well in low dimensional settings, their performance tends to degrade in high-dimensional settings that require large networks.

Gal & Ghahramani (2016) proved that dropout can also be used as a form of Bayesian Inference. Specifically, if a network is trained using dropout and L2 regularization, posterior samples can be collected by applying dropout at test-time. This method has surprisingly good out-of-distribution detection and can be applied to any network. However, it can be hard to tune the dropout parameter to have good uncertainty prediction without significantly degrading performance.

Neural network ensembling is another commonly used epistemic uncertainty estimation technique, which is closely related to our proposed method in this work. This technique involves training a number of completely separate neural networks on the same data to form an ensemble. Because of the stochasticity of initialization and the training process, each member of the ensemble should be a slightly different function. Specifically, the ensemble members should produce similar outputs in data-dense regions of the state-space and dissimilar outputs in data-sparse regions. Pearce et al. (2020) proved that, by using the correct regularization, each ensemble member becomes an approximate sample from the function posterior. Ensembling methods are particularly attractive to practitioners because they require very little fine-tuning and work surprisingly well in practice. However, these methods have one main drawback, namely that each new sample from the posterior

requires training a new network from scratch. Our method, on the other hand, seeks to construct a generative posterior model using similar regularization techniques to Pearce et al. (2020), allowing for quick sampling from the posterior.

# 3 PROBLEM STATEMENT AND BACKGROUND

We will focus on the problem of Bayesian Inference on the parameters of $\boldsymbol{\theta}$ of some function $f(\cdot; \boldsymbol{\theta})$. We assume we have some known prior over parameters $P_{\text{prior}}(\boldsymbol{\theta}) \sim \mathcal{N}(\mu_{\text{prior}}, \boldsymbol{\Sigma}_{\text{prior}})$ as well as access to a noisy dataset $(\mathbf{x}_{\text{obs}}, \mathbf{y}_{\text{obs}}) = \{(x_{\text{obs}}^1, y_{\text{obs}}^1) \ldots, (x_{\text{obs}}^N, y_{\text{obs}}^N)\}$, where $y_{\text{obs}}^i = f(x_{\text{obs}}^i; \boldsymbol{\theta}) + \epsilon$, $\epsilon \sim \mathcal{N}(0, \sigma_\epsilon)$ for some unknown parameter $\boldsymbol{\theta}$. We can then define the data-likelihood as

$$P_{\text{like}}(\mathbf{y}_{\text{obs}} \,|\, \boldsymbol{\theta}, \mathbf{x}_{\text{obs}}) = \prod_i \mathcal{N}(y_{\text{obs}}^i \,|\, f(x_{\text{obs}}^i; \boldsymbol{\theta}), \sigma_\epsilon). \tag{1}$$

The goal is to approximate samples from the posterior,

$$P_{\text{post}}(\boldsymbol{\theta} \,|\, \mathbf{y}_{\text{obs}}, \mathbf{x}_{\text{obs}}) \propto P_{\text{like}}(\mathbf{y}_{\text{obs}} \,|\, \boldsymbol{\theta}, \mathbf{x}_{\text{obs}}) P_{\text{prior}}(\boldsymbol{\theta}). \tag{2}$$

For ease of notation, we will drop the $\mathbf{x}_{\text{obs}}$ from the likelihood.

For our analysis, it is convenient if the posterior distribution is Gaussian. To show this, we will use the fact that the product of two Gaussian PDFs is proportional to a Gaussian PDF. We make a distinction here between the data-likelihood, a function of the observed data $\mathbf{y}_{\text{obs}}$, and the parameter-likelihood, a function of the parameters $\boldsymbol{\theta}$, defined as follows:

$$P_{\text{param-like}}(\boldsymbol{\theta}; \mathbf{y}_{\text{obs}}) = P_{\text{like}}(\mathbf{y}_{\text{obs}} \,|\, \boldsymbol{\theta}). \tag{3}$$

The parameter-likelihood PDF is equivalent to the data-likelihood PDF for all values of $\mathbf{y}_{\text{obs}}$ and $\boldsymbol{\theta}$, but is notably a function of the parameters $\boldsymbol{\theta}$. For a more detailed explanation of the distinction between the parameter and data-likelihoods, see Definition 1 from Pearce et al. (2020).

When the function $f$ is linear, it can be shown that the parameter-likelihood is also Gaussian,

$$P_{\text{param-like}}(\boldsymbol{\theta}; \mathbf{y}_{\text{obs}}) \propto \mathcal{N}(\boldsymbol{\mu}_{\text{like}}, \boldsymbol{\Sigma}_{\text{like}}). \tag{4}$$

While this is not true in general, Pearce et al. (2020) assume that the parameter likelihood is Gaussian so that the posterior also becomes Gaussian,

$$P_{\text{post}}(\boldsymbol{\theta} \,|\, \mathbf{y}_{\text{obs}}) \propto P_{\text{param-like}}(\boldsymbol{\theta}; \mathbf{y}_{\text{obs}}) P_{\text{prior}}(\boldsymbol{\theta}) \propto \mathcal{N}(\boldsymbol{\mu}_{\text{like}}, \boldsymbol{\Sigma}_{\text{like}}) \mathcal{N}(\boldsymbol{\mu}_{\text{prior}}, \boldsymbol{\Sigma}_{\text{prior}}) = \mathcal{N}(\boldsymbol{\mu}_{\text{post}}, \boldsymbol{\Sigma}_{\text{post}}), \tag{5}$$

where the posterior covariance and mean are given by

$$\boldsymbol{\Sigma}_{\text{post}} = \left( \boldsymbol{\Sigma}_{\text{like}}^{-1} + \boldsymbol{\Sigma}_{\text{prior}}^{-1} \right)^{-1}, \quad \boldsymbol{\mu}_{\text{post}} = \boldsymbol{\Sigma}_{\text{post}} \boldsymbol{\Sigma}_{\text{like}}^{-1} \boldsymbol{\mu}_{\text{like}} + \boldsymbol{\Sigma}_{\text{post}} \boldsymbol{\Sigma}_{\text{prior}}^{-1} \boldsymbol{\mu}_{\text{prior}}. \tag{6}$$

Note, however, that after the reformulation of the problem in Section 4.1, we no longer require this assumption for our theoretical analysis.

## 3.1 RANDOMIZED MAP SAMPLING (RMS)

Since the values $\boldsymbol{\mu}_{\text{like}}$ and $\boldsymbol{\Sigma}_{\text{like}}$ are intractible to compute in practice, the challenge of computing the posterior distribution remains. However, Pearce et al. (2020) showed how we can use a method called Randomized MAP sampling (RMS) to approximately sample from the posterior distribution. With RMS we first assume access to a mechanism to compute the maximum a posteriori (MAP) solution for our posterior. Then, by randomly shifting the prior distribution we use in our MAP solver, we can estimate samples from the posterior.

Specifically, RMS samples "anchor points" $\boldsymbol{\theta}_{\text{anc}}$ from some distribution $\boldsymbol{\theta}_{\text{anc}} \sim \mathcal{N}(\boldsymbol{\mu}_{\text{anc}}, \boldsymbol{\Sigma}_{\text{anc}})$. Each anchor point defines a shifted prior distribution $P_{\text{anc}}(\boldsymbol{\theta}_{\text{anc}}) = \mathcal{N}(\boldsymbol{\theta}_{\text{anc}}, \boldsymbol{\Sigma}_{\text{prior}})$. Now we define the MAP function which finds the MAP solution using this shifted prior:

$$\theta_{\text{MAP}}(\boldsymbol{\theta}_{\text{anc}}) = \arg\max_{\boldsymbol{\theta}} P_{\text{like}}(\mathbf{y}_{\text{obs}} \,|\, \boldsymbol{\theta}) P_{\text{anc}}(\boldsymbol{\theta}). \tag{7}$$

By sampling different anchor points, we can then construct a probability distribution over MAP solutions $P(\theta_{\text{MAP}}(\boldsymbol{\theta}_{\text{anc}}))$, which can be shown to be Gaussian. Using moment matching, Pearce et al. (2020) proved, in Theorem 1, that if we choose $\boldsymbol{\mu}_{\text{anc}} = \boldsymbol{\mu}_{\text{prior}}$ and $\boldsymbol{\Sigma}_{\text{anc}} = \boldsymbol{\Sigma}_{\text{prior}} + \boldsymbol{\Sigma}_{\text{prior}} \boldsymbol{\Sigma}_{\text{like}}^{-1} \boldsymbol{\Sigma}_{\text{prior}}$, then $P(\theta_{\text{MAP}}(\boldsymbol{\theta}_{\text{anc}})) = P_{\text{post}}(\boldsymbol{\theta} \,|\, \mathbf{y}_{\text{obs}})$. Using the approximation $\boldsymbol{\Sigma}_{\text{anc}} \approx \boldsymbol{\Sigma}_{\text{prior}}$, a sample from posterior, $\boldsymbol{\theta}_{\text{post}}$, can be collected by sampling $\boldsymbol{\theta}_{\text{anc}} \sim \mathcal{N}(\boldsymbol{\mu}_{\text{anc}}, \boldsymbol{\Sigma}_{\text{anc}})$ and optimizing the following:

$$\boldsymbol{\theta}_{\text{post}} = \arg\max_{\boldsymbol{\theta}} \log P_{\text{param-like}}(\boldsymbol{\theta}; \mathbf{y}_{\text{obs}}) + \log P_{\text{anc}}(\boldsymbol{\theta}). \tag{8}$$

## 4 GENERATIVE POSTERIOR NETWORKS

While RMS can be used to construct samples from the posterior, every new sample requires solving a new optimization problem. Instead, we propose learning the MAP function itself.

Let $g$ be a neural network parameterized by some vector, $\phi$, that takes as input a sample from the anchor distribution, $\boldsymbol{\theta}_{\text{anc}} \sim P_{\text{anc}}(\boldsymbol{\theta}_{\text{anc}})$, and outputs $\theta_{\text{MAP}}(\boldsymbol{\theta}_{\text{anc}})$. If we assume that $g$ has enough expressive power to represent the MAP function, then $g(\boldsymbol{\theta}_{\text{anc}}; \phi) = \theta_{\text{MAP}}(\boldsymbol{\theta}_{\text{anc}})$ if:

$$\phi = \arg\max_{\phi} \mathbb{E}_{\boldsymbol{\theta}_{\text{anc}} \sim P_{\text{anc}}(\boldsymbol{\theta}_{\text{anc}})} \log P_{\text{like}}(\mathbf{y}_{\text{obs}} \mid g(\boldsymbol{\theta}_{\text{anc}}; \phi)) + \log P_{\text{anc}}(g(\boldsymbol{\theta}_{\text{anc}}; \phi)). \tag{9}$$

In practice, however, this is a very hard optimization problem for two main reasons. (1) Optimizing in parameter space tends to be challenging for gradient-descent optimization. (2) The space of all parameters $\boldsymbol{\theta}_{\text{anc}}$ is not only very large, but there are also many parameters $\boldsymbol{\theta}_{\text{anc}}$ that map to the same outputs for all inputs. Moreover, we would expect the posterior parameters to be highly correlated and, therefore, need less expressive power to be represented.

To address these two problems, we will first change the problem definition slightly to find the posterior in *output* space as opposed to *parameter* space. We prove in the next section that, by changing our loss function, we can still use RMS to approximate the posterior in output space. Next, we show practically how we construct a low-dimensional representation of the anchor distribution.

### 4.1 THEORETICAL RESULTS

As mentioned above, optimizing a generative model to output parameters can be challenging. Instead, we reformulate the Bayesian Inference problem slightly and consider the posterior in output space. Concretely, consider some discretization of the input space, $\mathbf{x}_{\text{sample}} = \{x^1_{\text{sample}}, \ldots, x^M_{\text{sample}}\}$. Let $\hat{Y}_i = f(x^i_{\text{sample}}; \boldsymbol{\theta})$ be a set of sample points and $\hat{\mathbf{Y}} = \{\hat{Y}_1, \ldots, \hat{Y}_M\}$ be a transformation of the random variable $\boldsymbol{\theta}$, where each element is the output of the function $f$ parameterized by $\boldsymbol{\theta}$ evaluated at points $\mathbf{x}_{\text{sample}}$. The prior and likelihood for this transformed random variable $\hat{\mathbf{Y}}$ can be expressed in terms of the prior and likelihood of $\boldsymbol{\theta}$:

$$P(\hat{\mathbf{Y}}) = \int_{\boldsymbol{\theta}} \mathbf{1}_{\hat{Y}_i = f(x^i_{\text{sample}}; \boldsymbol{\theta}) \forall i} P_{\text{prior}}(\boldsymbol{\theta}) d\boldsymbol{\theta}, \tag{10}$$

$$P(\mathbf{y}_{\text{obs}} \mid \hat{\mathbf{Y}}) \propto \int_{\boldsymbol{\theta}} \mathbf{1}_{\hat{Y}_i = f(x^i_{\text{sample}}; \boldsymbol{\theta}) \forall i} P_{\text{like}}(\mathbf{y}_{\text{obs}} \mid \boldsymbol{\theta}) P_{\text{prior}}(\boldsymbol{\theta}) d\boldsymbol{\theta}. \tag{11}$$

Note that samples from the prior of $\hat{\mathbf{Y}}$ can be easily obtained by sampling $\boldsymbol{\theta} \sim P_{\text{like}}(\boldsymbol{\theta})$ and computing $\hat{Y}_i = f(x^i_{\text{sample}}; \boldsymbol{\theta})$ for all $i$. The goal in this reformulated problem is to find the posterior $P(\hat{\mathbf{Y}} \mid \mathbf{y}_{\text{obs}}) \propto P(\mathbf{y}_{\text{obs}} \mid \hat{\mathbf{Y}}) P(\hat{\mathbf{Y}})$.

As before, we estimate the posterior using RMS and define a new MAP function in the output space:

$$\hat{Y}_{\text{MAP}}(\hat{\mathbf{Y}}_{\text{anc}}) = \arg\max_{\hat{\mathbf{Y}}} P(\mathbf{y}_{\text{obs}} \mid \hat{\mathbf{Y}}) P_{\text{anc}}(\hat{\mathbf{Y}}_{\text{anc}}) \tag{12}$$

for some anchor distribution $\hat{\mathbf{Y}}_{\text{anc}} \sim P_{\text{anc}}(\hat{\mathbf{Y}}_{\text{anc}}) = \mathcal{N}(\boldsymbol{\mu}_{\text{anc}}, \boldsymbol{\Sigma}_{\text{anc}})$. We now need to show that, with the correct choice of $\boldsymbol{\mu}_{\text{anc}}$ and $\boldsymbol{\Sigma}_{\text{anc}}$, $P(\hat{Y}_{\text{MAP}}(\hat{\mathbf{Y}}_{\text{anc}})) = P(\hat{\mathbf{Y}} \mid \mathbf{y}_{\text{obs}})$.

To simplify the analysis, we assume that the output prior is normally distributed $P(\hat{\mathbf{Y}}) = \mathcal{N}(\boldsymbol{\mu}_{\hat{\mathbf{Y}}}, \boldsymbol{\Sigma}_{\hat{\mathbf{Y}}})$. Since the parameters are normally distributed, this approximation becomes exact when either $f$ is linear or the final layer is infinitely wide.

Next, we need to show that the likelihood function $P(\mathbf{y}_{\text{obs}} \mid \hat{\mathbf{Y}})$ is roughly normally distributed with respect to the vector $\hat{\mathbf{Y}}$. Recall that each observed data point, $y^i_{\text{obs}}$, is equal to a noisy output of $f$ for some parameter $\boldsymbol{\theta}$, $y^i_{\text{obs}} = f(x^i_{\text{obs}}; \boldsymbol{\theta}) + \epsilon$. In the case where $\mathbf{x}_{\text{obs}} \subset \mathbf{x}_{\text{sample}}$, $y^i_{\text{obs}}$ is evaluated on the same points as $\hat{\mathbf{Y}}$, so the likelihood of $P(\mathbf{y}_{\text{obs}} \mid \boldsymbol{\theta})$ becomes equivalent to $P(\mathbf{y}_{\text{obs}} \mid \hat{\mathbf{Y}})$. By the same reasoning, if we choose $\mathbf{x}_{\text{sample}}$ to contain all the points in the domain $\mathcal{X}$, $P(\mathbf{y}_{\text{obs}} \mid \boldsymbol{\theta})$ becomes equivalent to $P(\mathbf{y}_{\text{obs}} \mid \hat{\mathbf{Y}})$.

**Lemma 1.** *Let $f$ be some neural network parameterized by $\boldsymbol{\theta}$. Let $\mathbf{x}_{obs}, \mathbf{y}_{obs}$ correspond to vectors of observed labeled data. Let $\mathbf{x}_{sample}$ be some discretization over the input space of $M^d$ points such that for any $x \in \mathcal{X}$ there exists an $x' \in \mathbf{x}_{sample}$ where $\|x - x'\|_2 \leq \frac{l}{M}$ for some constant $l$. Define $\hat{\mathbf{Y}} := \{f(x; \boldsymbol{\theta}) : \forall x \in \mathbf{x}_{sample}\}$ be a vector of evaluations of $f$ on all points in $\mathbf{x}_{sample}$ for a given parameter vector $\boldsymbol{\theta}$.*

*Then, if we take the limit as $M \to \infty$, $P(\mathbf{y}_{obs} \,|\, \hat{\mathbf{Y}}) = P(\mathbf{y}_{obs} \,|\, \boldsymbol{\theta})$.*

*Proof.* See Appendix A.1. $\qquad\qquad\qquad\qquad\qquad\qquad\qquad\qquad\qquad\qquad\qquad\qquad$ $\square$

It turns out that, since the observation noise is additive, the likelihood in parameter space $P(\mathbf{y}_{\text{obs}} \,|\, \boldsymbol{\theta})$ is also Gaussian as a function of the output vector $\hat{\mathbf{Y}}$. Thus, if we take $|\mathbf{x}_{\text{sample}}| \to \infty$ and apply Lemma 1, we get that the output likelihood $P(\mathbf{y}_{\text{obs}} \,|\, \hat{\mathbf{Y}})$ is Gaussian as a function of $\hat{\mathbf{Y}}$.

Now, since we have that both the prior and likelihood of $\hat{\mathbf{Y}}$ can be expressed as Gaussian functions of $\hat{\mathbf{Y}}$, we also have that the posterior is Gaussian. As before, we just need to show that the distribution of MAP solutions $P(\hat{Y}_{\text{MAP}}(\hat{\mathbf{Y}}_{\text{anc}}))$ is also Gaussian, then apply moment matching to find the right values of $\boldsymbol{\mu}_{\text{anc}}$ and $\boldsymbol{\Sigma}_{\text{anc}}$ such that $P(\hat{Y}_{\text{MAP}}(\hat{\mathbf{Y}}_{\text{anc}}))) = P(\hat{\mathbf{Y}} \,|\, \mathbf{y}_{\text{obs}})$.

**Theorem 1.** *Let $f$ be some neural network parameterized by $\boldsymbol{\theta}$. Let $\mathbf{x}_{sample}$ be some vector of inputs sampled uniformly from $\mathcal{X}^M$. Define $\hat{\mathbf{Y}} := [f(x_1; \boldsymbol{\theta}), \dots, f(x_M; \boldsymbol{\theta})]$ as a transformation of the random variable $\boldsymbol{\theta}$. Assume the prior of $\hat{\mathbf{Y}}$ follows a multivariate normal distribution, $P(\hat{\mathbf{Y}}) = \mathcal{N}(\boldsymbol{\mu}_{\hat{\mathbf{Y}}}, \boldsymbol{\Sigma}_{\hat{\mathbf{Y}}})$.*

*We assume access to some function which takes as input the center of a shifted distribution and outputs the MAP estimate of $\hat{\mathbf{Y}}$, $\hat{Y}_{MAP}(\hat{\mathbf{Y}}_{anc})$.*

*If we choose the distribution over $\hat{\mathbf{Y}}_{anc}$ to be $P(\hat{\mathbf{Y}}_{anc}) = \mathcal{N}(\boldsymbol{\mu}_{anc}, \boldsymbol{\Sigma}_{anc})$, where $\boldsymbol{\mu}_{anc} = \boldsymbol{\mu}_{\hat{\mathbf{Y}}}$ and $\boldsymbol{\Sigma}_{anc} = \boldsymbol{\Sigma}_{\hat{\mathbf{Y}}} + \boldsymbol{\Sigma}_{\hat{\mathbf{Y}}} \boldsymbol{\Sigma}_{like}^{-1} \boldsymbol{\Sigma}_{\hat{\mathbf{Y}}}$, then $\lim_{M \to \infty} P(\hat{Y}_{MAP}(\hat{\mathbf{Y}}_{anc})) = P(\hat{\mathbf{Y}} \,|\, \mathbf{y}_{obs})$.*

*Proof.* See Appendix A.2. $\qquad\qquad\qquad\qquad\qquad\qquad\qquad\qquad\qquad\qquad\qquad\qquad$ $\square$

## 4.2 Practical Implementation

In practice, as was done in Pearce et al. (2020), we will instead use the approximation $\boldsymbol{\Sigma}_{\text{anc}} \approx \boldsymbol{\Sigma}_{\hat{\mathbf{Y}}}$ to make it tractable to sample from the anchor distribution. Pearce et al. (2020) argue that this approximation, in general, causes RMS to over-estimate the posterior variance.

With this assumption, the anchor distribution $P_{\text{anc}}(\hat{\mathbf{Y}}_{\text{anc}})$ becomes approximately equal to the prior distribution $P(\hat{\mathbf{Y}})$. Recall that, to sample from the output prior, we simply sample a set of parameters $\boldsymbol{\theta} \sim P_{\text{prior}}(\boldsymbol{\theta})$ and evaluate the function $f$ at all evaluation points $\mathbf{x}_{\text{sample}}$. Now, if we want to learn the MAP function in the output space, we can construct a neural network $g$, parameterized by some vector $\phi$, that takes a sample from the output anchor distribution, $\hat{\mathbf{Y}}_{\text{anc}} \sim \mathcal{N}(\boldsymbol{\mu}_{\text{anc}}, \boldsymbol{\Sigma}_{\text{anc}})$, along with the evaluation points $\mathbf{x}_{\text{sample}}$, and outputs $\theta_{\text{MAP}}(\boldsymbol{\theta}_{\text{anc}})$. If we assume that $g$ has enough expressive power to represent the MAP function, then $g(\hat{\mathbf{Y}}_{\text{anc}}; \phi) = \hat{Y}_{\text{MAP}}(\hat{\mathbf{Y}}_{\text{anc}})$, if:

$$\phi = \arg\max_{\phi} \mathbb{E}_{\boldsymbol{\theta}_{\text{anc}} \sim P_{\text{prior}}(\boldsymbol{\theta}_{\text{anc}})} \log P_{\boldsymbol{\theta}}(\mathbf{y}_{\text{obs}} \,|\, g(\mathbf{x}_{\text{sample}}, \boldsymbol{\theta}_{\text{anc}}; \phi)) + \log P_{\text{anc}}(g(\mathbf{x}_{\text{sample}}, \boldsymbol{\theta}_{\text{anc}}; \phi)) \quad (13)$$

This can be re-written as:

$$\phi = \arg\min_{\phi} \mathbb{E}_{\boldsymbol{\theta}_{\text{anc}} \sim P_{\text{prior}}(\boldsymbol{\theta}_{\text{anc}})} \sum_{i=1}^{N} \|y_{\text{obs}}^i - g(x_{\text{obs}}^i, \boldsymbol{\theta}_{\text{anc}}; \phi)\|_2^2 + \sigma_\epsilon^2 \boldsymbol{\delta}^T \boldsymbol{\Sigma}_{\hat{\mathbf{Y}}}^{-1} \boldsymbol{\delta}, \quad (14)$$

where $\delta_j = g(x_{\text{sample}}^j, \boldsymbol{\theta}_{\text{anc}}; \phi) - f(x_{\text{sample}}^j; \boldsymbol{\theta}_{\text{anc}})$. See Appendix A.3 for a step-by-step derivation.

In practice, of course, we cannot compute the regularization term, $\frac{1}{N} \sigma_\epsilon^2 \boldsymbol{\delta}^T \boldsymbol{\Sigma}_{\hat{\mathbf{Y}}}^{-1} \boldsymbol{\delta}$ as $|\mathbf{x}_{\text{sample}}| \to \infty$. Instead, we approximate this term using a finite number of sample points from the input space $\mathcal{X}$. Because of this, for high-dimensional problems, the off-diagonal terms of $\boldsymbol{\Sigma}_{\hat{\mathbf{Y}}}$ are near zero almost

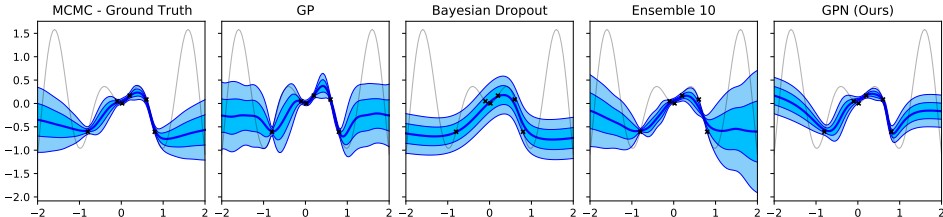

Figure 2: Predicted posterior distributions of different methods using the same observed data. Empirical distributions are constructed using 100 samples for all methods except for the Ensemble-10 method, which is only over 10 ensemble members.

always. However, we found that, in practice, assuming that the samples of $\hat{\mathbf{Y}}$ are independent did not significantly degrade performance. Using this approximation, our optimization problem becomes:

$$\phi = \arg\min_{\phi} \mathbb{E}_{\boldsymbol{\theta}_{\text{anc}} \sim P_{\text{prior}}(\boldsymbol{\theta}_{\text{anc}}), x_{\text{sample}} \sim \text{Unif}(\mathcal{X})} \sum_{i=1}^{N} \|y_{\text{obs}}^i - g(x_{\text{obs}}^i, \boldsymbol{\theta}_{\text{anc}}; \phi)\|_2^2 + \beta\|\delta\|_2^2, \quad (15)$$

where $\delta = g(x_{\text{sample}}, \boldsymbol{\theta}_{\text{anc}}; \phi) - f(x_{\text{sample}}; \boldsymbol{\theta}_{\text{anc}})$ and $\beta$ is a hyper-parameter.

In order for $g$ to have enough expressive power to represent the full MAP function, it may require the size of the parameters $\phi$ to be much larger than $\boldsymbol{\theta}$. However, after applying the low-dimensional approximation of $\boldsymbol{\theta}_{\text{anc}}$ described below, in our experiments, we only used roughly double the number of parameters compared to each member of the ensemble method. And since the ensemble we used has 10 members, the total number of parameters needed for our method is significantly smaller.

## 4.3 LOW-DIMENSIONAL EMBEDDING

As mentioned above, optimizing the generative function $g$ over the the space of all anchor parameters $\boldsymbol{\theta}_{\text{anc}}$ did not yield great results because the space was too large. Instead, we sought to learn a low-dimensional representation of the anchor parameters. There are two key reasons why we would expect to be able to decrease the representational power of the anchor distribution and maintain the same of fidelity in the posterior estimation. (1) Because of the nature of neural networks, there are many settings of parameters $\boldsymbol{\theta}$ that would result in the same output vector $\hat{\mathbf{Y}}$. Because we are focusing on the posterior of $\hat{\mathbf{Y}}$, we need less representational power to model $\hat{\mathbf{Y}}$. (2) And, maybe more importantly, because the posterior parameters are highly correlated, we would expect to need less expressive power to represent the posterior distribution than the prior distribution.

Thus, we would like to construct a simpler estimate of the prior space using a low-dimensional embedding vector, $\mathbf{z}$. That is, we want our generative model to take as input the embedding vector $\mathbf{z} \sim \mathcal{N}(0, I)$, instead of $\boldsymbol{\theta}_{\text{anc}}$, in order to estimate the MAP function. To do this, we need to learn a mapping from anchor parameters $\boldsymbol{\theta}_{\text{anc}}$ to embedding vectors $\mathbf{z}$. For our experiments, we used a 1-1 embedding scheme where we sample $k$ parameters from the true prior $\boldsymbol{\theta}_1, \ldots, \boldsymbol{\theta}_k \sim P_{\text{prior}}(\boldsymbol{\theta})$ and $k$ independent samples from our embedding $\mathbf{z}_1, \ldots, \mathbf{z}_k$. We then jointly optimize over $\phi$ and $\mathbf{z}_1, \ldots, \mathbf{z}_k$ as follows:

$$\arg\min_{\phi, \mathbf{z}_1, \ldots, \mathbf{z}_k} \mathbb{E}_{j \in [1, \ldots, k]} \sum_{i=1}^{N} \|y_{\text{obs}}^i - g(x_{\text{obs}}^i, \mathbf{z}_j + \boldsymbol{\epsilon}; \phi)\|_2^2 + \beta\|\delta\|_2^2 + \mathcal{L}_{\text{reg}}(\mathbf{z}_1, \ldots, \mathbf{z}_k), \quad (16)$$

where $\delta_j = g(x_{\text{sample}}^j, \mathbf{z}_j + \boldsymbol{\epsilon}; \phi) - f(x_{\text{sample}}^j; \boldsymbol{\theta}_j)$, $\boldsymbol{\epsilon} \sim \mathcal{N}(0, I)$ is a noise injection vector that improves the smoothness of interpolations between embedding vectors, and $\mathcal{L}_{\text{reg}}(\mathbf{z}_1, \ldots, \mathbf{z}_k)$ is a regularizer to keep $\mathbf{z}_1, \ldots, \mathbf{z}_k$ roughly normally distributed, which allows us to easily sample from the embedding space.. For our experiments we use the KL divergence between $\mathbf{z}$ and the normal distribution, $\mathcal{L}_{\text{reg}}(\mathbf{z}_1, \ldots, \mathbf{z}_k) = D_{\text{KL}}(\mathcal{N}(\bar{\mathbf{z}}, s_{\mathbf{z}}), \mathcal{N}(0, 1))$. Figure 1 illustrates how we can sample from this embedding space to construct posterior functions.

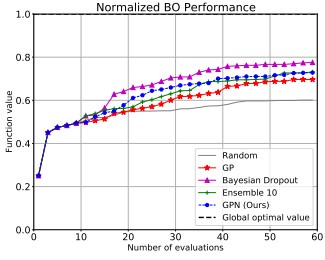

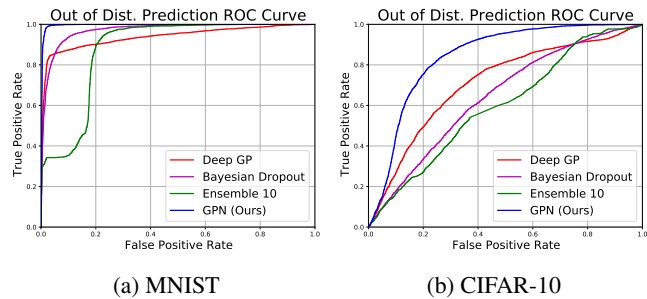

Figure 3: Best normalized sample value for each method throughout training averaged across 5 BO problems and 5 trials for each problem.

(a) MNIST       (b) CIFAR-10

Figure 4: ROC curves for out of distribution prediction based on sample variance from 100 samples.

## 4.4 CLASSIFICATION

Our method uses the assumption that the prior over $\hat{\mathbf{Y}}$ is approximately normally distributed. While this approximation is exact, assuming the prior parameters are normally distributed, only when either the network is linear or the final layer is infinitely wide and does not have an activation function, for most deep networks, this is a good approximation. However, if we add a soft-max to the final output, as is usually done for classification tasks, this assumption is violated. To get around this, for classification tasks, we add the anchor loss to the pre-softmax outputs.

## 5 EXPERIMENTS

The goal of our experiments is to illustrate the ability of our method to accurately model epistemic uncertainty for out of distribution data while retaining high performance on in distribution data. We will test our method on three problems: (1) a small-scale regression task, (2) Bayesian Optimization (BO), and (3) Classification with out of distribution examples.

We compare our method against 3 competing Bayesian methods: Gaussian Processes (GPs), Bayesian Dropout (Gal & Ghahramani, 2016), and anchor-regularized neural-network ensembles (Pearce et al., 2020) with 10 ensemble members. For our BO experiments we use exact GP with a static kernel since the problems are sufficiently low-dimensional, but for our classification experiments we need to use approximate GPs. Specifically, we use a grid-interpolated GP with Deep Kernel Learning (Wilson et al., 2016) implemented with the GPyTorch library (Gardner et al., 2018). For the anchor-regularized ensembles, since every member requires a unique set of parameters (and anchor points) the number of ensemble members was limited by the memory capacity of our GPUs.

## 5.1 SMALL SCALE REGRESSION

Our first experiment is a small 1-dimensional regression problem where we can easily compute the ground truth. We provide each method with the same 6 observations. For this experiment, we compute 100 samples from the approximate posterior for each method (except for the Ensemble where we only use 10 samples). For ground truth, we use the Metropolis-Hastings MCMC algorithm. Figure 2 shows the results.

## 5.2 BAYESIAN OPTIMIZATION

The goal of Bayesian Optimization is to find the maximum of a function in as few samples as possible. This is commonly done by first constructing a posterior over possible functions given the samples we have collected so far, then, through various methods, choosing sample points that might maximize the true function. By using different methods for predicting the posterior in BO, we can evaluate how well these methods capture epistemic uncertainty.

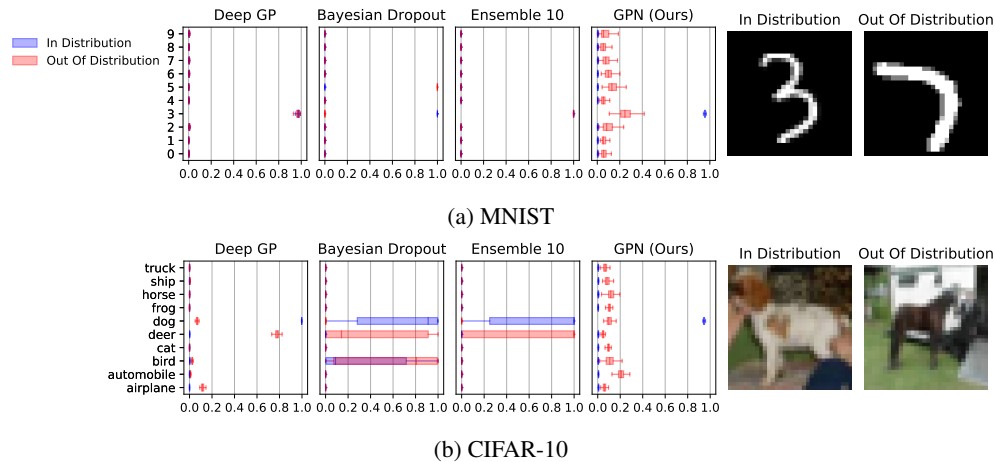

(a) MNIST

(b) CIFAR-10

Figure 5: Boxplots of 100 posterior samples from every method for 2 test images, one from a class that the model was trained on ("In Distribution") and one from a class that the model wasn't trained on ("Out Of Distribution").

Table 1: Classification performance on both the MNIST and CIFAR-10 datasets for In Distribution and Out of Distribution examples. CI-Any refers to the proportion of data points for which *at least one* sample (out of 100) contains the correct label and CI-All refers to the proportion of data points for which *every* sample predicts the correct label.

| Dataset | | | Deep GP | Dropout | Ensemble 10 | GPN (Ours) |
|---|---|---|---|---|---|---|
| MNIST | In Dist. | Accuracy | **99.7**% | **99.2**% | **99.0**% | **99.5**% |
| | | CI-All | **98.9**% | **89.8**% | 0.0% | **97.9**% |
| | Out of Dist. | CI-Any | 0.0% | 9.4% | 0.0% | **79.1**% |
| | Out of Dist. ROC AUC | | 0.94 | 0.97 | 0.88 | **0.99** |
| CIFAR-10 | In Dist. | Accuracy | **88.7**% | 82.3% | **87.3**% | **88.5**% |
| | | CI-All | **87.0**% | 68.5% | 0.0% | **87.2**% |
| | Out of Dist. | CI-Any | 0.0% | 0.0% | 0.0% | **67.6**% |
| | Out of Dist. ROC AUC | | 0.71 | 0.64 | 0.60 | **0.84** |

For all methods we use Thompson Sampling and implemented the experiments using the BOTorch library by Balandat et al. (2020). We tested each method on five standard benchmark BO functions: the Branin, Hartmann-6D, Shekel, Michalewicz-10D, and Powell functions. We normalize the output of each function between a minimum value (found by random sampling) and the optimal value. To make sure that each method had a large enough sample variance, we chose hyper-parameters such that 100 prior samples from each method have a range of roughly $[-1, 1]$. To choose the best regularization hyper-parameter, we ran each method with 5 different regularization parameters, then chose the hyper-parameters with the highest average performance across all BO problems.

Figure 3 shows the best normalized function value sampled so far during optimization averaged across the 5 BO problems each ran for 5 trials. Our method significantly out-performs Random sampling and even slightly outperforms GPs, one of the best BO techniques for low-dimensional problems. Even though our method trails slightly behind Bayesian Dropout, these results still illustrate the uncertainty estimation power of our method, even on low-dimensional problems.

## 5.3 CLASSIFICATION

We also look at the problem of image classification. Along with our method having high accuracy, we are also interested in how well our method (and the baseline methods) is able to capture epistemic uncertainty. To measure this we split the dataset into two sets based on class labels: the in distribution dataset and the out of distribution dataset. During training, we only provide the labels for

the in-distribution dataset. During testing, along with recording the accuracy on the in distribution dataset, we also sample our models on the out of distribution dataset. For such out of distribution examples, we expect the models to show high epistemic uncertainty and, thus, the samples for these out of distribution examples should have high variance.

Note that we still provide the full set of unlabeled training data to our agent. This obviously favors our method over the baseline methods as our method is the only method that is able to take advantage of this unlabeled data. However, from a practical perspective this is a reasonable setup as many real-world datasets have far more unlabeled data than labeled data.

One desired property of an epistemic uncertainty estimator is in constructing confidence sets. For the classification setting, the confidence interval (CI) we consider is the set of possible class predictions. We would expect that this confidence interval contains the true class with high probability. We also expect that, on in-distribution examples, the confidence interval *only* contains the true label with high probability. To construct these confidence intervals, we take 100 samples from each model on every data point in the test data. We define the metrics CI-Any as the proportion of data points for which *at least one* sample contains the correct label and CI-All as the proportion of data points for which *every* sample predicts the correct label.

Another desired property of an epistemic uncertainty estimator is in predicting when a data point is outside the training distribution and, thus, any classification prediction made on such a data point will likely be incorrect. For out of distribution data points, we expect high sample variance from the posterior model and low sample variance from in distribution data points. Thus, variance can be used as a metric to predict which test data points are outside the training distribution. To quantify each method's ability to perform out of distribution detection using sample variance, we construct ROC curves for each method and measure the area under the curve (AUC).

We tested all methods on both the MNIST dataset and the CIFAR-10 dataset. For MNIST, the in distribtuion classes were $[0, 1, 2, 3, 4, 5]$ and the out of distribution classes were $[6, 7, 8, 9]$. For CIFAR-10, the in distribtuion classes were ["airplane", "automobile", "bird", "deer", "dog", "ship"] and the out-of-distribution classes were ["cat", "frog", "horse", "truck"]. To make sure that each method had a large enough sample variance, we chose hyper-parameters such that prior distributions had near 100% CI-All. To choose the best regularization hyper-parameter, as in the BO experiments, we ran each method with 5 different regularization parameters, then chose the model with both a high validation accuracy and CI-All.

For all other methods, we take 100 samples for measuring CI-Any, CI-All, and sample variance.

Table 1 summarizes the performance of each model using the metrics of test accuracy, CI-Any, CI-All, and the ROC AUC for out of distribution detection. Full ROC curves for each method are plotted in Figure 4. While all methods are able to achieve high prediction accuracy and high *CI-All* on *in distribution* examples, our method is the only one to achieve high *CI-Any* on *out of distribution* examples, illustrating our method's ability to construct valid confidence intervals. Additionally, our method significantly out performs all other methods on out of distribution detection.

Figure 5 shows two informative test images, one from the in distribution dataset and one from the out of distribution dataset, for both MNIST and CIFAR-10, along with box plots of sampled PMFs from each of the learned models. While all methods predict the correct label with high precision on the in distribution example, ours has a uniquely wide prediction distribution on the out of distribution example, illustrating high predicted epistemic uncertainty.

## 6 CONCLUSION

In this paper we introduce Generative Posterior Networks (GPNs), a method to learn a generative model of the Bayesian posterior distribution by regularizing the outputs of the network towards samples of the prior. We prove that under mild assumptions, GPNs approximate samples from the true posterior. We then show empirically that our method is not only competitive with the best uncertainty predictions techniques on small Bayesian Optimization problems, it significantly outperforms these competing methods on high-dimensional classification tasks.

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

# A  PROOFS

## A.1  PROOF OF LEMMA 1

*Proof.* It is a standard result that linear feed-forward neural networks have bounded Lipschitz constants. We'll denote $L(\boldsymbol{\theta})$ as the Lipschitz constant of $f$ for a given set of parameters $\boldsymbol{\theta}$.

Recall that this likelihood is defined as follows:

$$P(\mathbf{y}_{\text{obs}} \mid \hat{\mathbf{Y}} \propto \int_{\boldsymbol{\theta}} \mathbf{1}_{\hat{Y}_i = f(x_{\text{sample}}^i; \boldsymbol{\theta}) \forall i} \left( \prod_i \mathcal{N}(y_{\text{obs}}^i \mid f(x_{\text{obs}}^i; \boldsymbol{\theta}), \sigma_\epsilon) \right) P_{\text{prior}}(\boldsymbol{\theta}) d\boldsymbol{\theta}$$

Let $\mathbf{x}_{\text{sample}}' \subset \mathbf{x}_{\text{sample}}$ be the subset of points in our discretization, $\mathbf{x}_{\text{sample}}$, that are closest to the observed points, $\mathbf{x}_{\text{obs}}$; in other words, for each element $x_{\text{obs}}^i \in \mathbf{x}_{\text{obs}}$, there exists an element $x_i \in \mathbf{x}_{\text{sample}}'$ where:

$$x_i' = \min_{x \in \mathbf{x}_{\text{sample}}} \|x_{\text{obs}}^i - x\|_2$$

We denote the elements of $\mathbf{x}_{\text{sample}}'$ as $[x_1', \ldots, x_N']$. Let $\hat{\mathbf{Y}}' \subset \hat{\mathbf{Y}}$ be the corresponding elements in $\hat{\mathbf{Y}}$; in other words, $\hat{Y}_i' = f(x_i'; \boldsymbol{\theta})$ for $i \in [1, \ldots, N]$.

By definition of our discretiztaion, we know that for all $i \in [1, \ldots, N]$, $\|x_{\text{obs}}^i - x_i'\| \leq \frac{l}{M}$. Using the Lipschitz-continuity of $f$, for any setting of parameters $\boldsymbol{\theta}$, $\|f(x_{\text{obs}}^i; \boldsymbol{\theta}) - \hat{Y}_i'\| \leq \frac{l}{M} L(\boldsymbol{\theta})$.

If we take the limit as $M \to \infty$, $f(x_{\text{obs}}^i; \boldsymbol{\theta}) = \hat{Y}_i$ and

$$P(\mathbf{y}_{\text{obs}} \mid \hat{\mathbf{Y}}) \propto \int_{\boldsymbol{\theta}} \mathbf{1}_{\hat{Y}_i = f(x_{\text{sample}}^i; \boldsymbol{\theta}) \forall i} \left( \prod_i \mathcal{N}(y_{\text{obs}}^i \mid \hat{Y}_i', \sigma_\epsilon) \right) P_{\text{prior}}(\boldsymbol{\theta}) d\boldsymbol{\theta}$$

$$= \left( \prod_i \mathcal{N}(y_{\text{obs}}^i \mid \hat{Y}_i', \sigma_\epsilon) \right) \int_{\boldsymbol{\theta}} \mathbf{1}_{\hat{Y}_i = f(x_{\text{sample}}^i; \boldsymbol{\theta}) \forall i} P_{\text{prior}}(\boldsymbol{\theta}) d\boldsymbol{\theta}$$

$$= \left( \prod_i \mathcal{N}(y_{\text{obs}}^i \mid \hat{Y}_i', \sigma_\epsilon) \right) P(\hat{\mathbf{Y}})$$

$$= \left( \prod_i \mathcal{N}(y_{\text{obs}}^i \mid f(x_{\text{obs}}^i; \boldsymbol{\theta}), \sigma_\epsilon) \right) P(\hat{\mathbf{Y}})$$

$$\propto P(\mathbf{y}_{\text{obs}} \mid \boldsymbol{\theta})$$

Now we know that $P(\mathbf{y}_{\text{obs}} \mid \hat{\mathbf{Y}}) \propto P(\mathbf{y}_{\text{obs}} \mid \boldsymbol{\theta})$. But since both $P(\mathbf{y}_{\text{obs}} \mid \hat{\mathbf{Y}})$ and $P(\mathbf{y}_{\text{obs}} \mid \boldsymbol{\theta})$ are both probability distributions of $\mathbf{y}_{\text{obs}}$, then $P(\mathbf{y}_{\text{obs}} \mid \hat{\mathbf{Y}}) = P(\mathbf{y}_{\text{obs}} \mid \boldsymbol{\theta})$.

$\square$

## A.2  PROOF OF THEOREM 1

*Proof.* We will start by showing that $P(\hat{\mathbf{Y}} \mid \mathbf{y}_{\text{obs}})$ is normally distributed. By Lemma 1, we know that $\lim_{M \to \infty} P(\mathbf{y}_{\text{obs}} \mid \hat{\mathbf{Y}}) = P(\mathbf{y}_{\text{obs}} \mid \hat{\mathbf{Y}})$. Thus, in the limit as $M \to \infty$,

$$P(\hat{\mathbf{Y}} \mid \mathbf{y}_{\text{obs}}) \propto P(\mathbf{y}_{\text{obs}} \mid \hat{\mathbf{Y}}) P(\hat{\mathbf{Y}})$$

$$= P(\mathbf{y}_{\text{obs}} \mid \boldsymbol{\theta}) P(\hat{\mathbf{Y}})$$

$$= \left( \prod_i \mathcal{N}(y_{\text{obs}}^i \mid f(x_{\text{obs}}^i; \boldsymbol{\theta}), \sigma_\epsilon) \right) P(\hat{\mathbf{Y}})$$

$$= \left( \prod_i \mathcal{N}(f(x_{\text{obs}}^i; \boldsymbol{\theta}) \mid y_{\text{obs}}^i, \sigma_\epsilon) \right) P(\hat{\mathbf{Y}})$$

$$= \mathcal{N}(\hat{\mathbf{Y}} \mid \boldsymbol{\mu}_{\text{like}}, \boldsymbol{\Sigma}_{\text{like}}) \mathcal{N}(\hat{\mathbf{Y}} \mid \boldsymbol{\mu}_{\hat{\mathbf{Y}}}, \boldsymbol{\Sigma}_{\hat{\mathbf{Y}}})$$

for some mean and covariance $\boldsymbol{\mu}_{\text{like}}, \boldsymbol{\Sigma}_{\text{like}}$. Since the product of two multivariate Gaussians is a Gaussian, we know that:

$$P(\hat{\mathbf{Y}} \,|\, \mathbf{y}_{\text{obs}}) = \mathcal{N}(\boldsymbol{\mu}_{\text{post}}, \boldsymbol{\Sigma}_{\text{post}}) \tag{17}$$

where

$$\boldsymbol{\Sigma}_{\text{post}} = \left( \boldsymbol{\Sigma}_{\text{like}}^{-1} + \boldsymbol{\Sigma}_{\hat{\mathbf{Y}}}^{-1} \right)^{-1}, \quad \boldsymbol{\mu}_{\text{post}} = \boldsymbol{\Sigma}_{\text{post}} \boldsymbol{\Sigma}_{\text{like}}^{-1} \boldsymbol{\mu}_{\text{like}} + \boldsymbol{\Sigma}_{\text{post}} \boldsymbol{\Sigma}_{\hat{\mathbf{Y}}}^{-1} \mu_{\hat{\mathbf{Y}}}. \tag{18}$$

Now we consider at the distribution $P(\hat{Y}_{\text{MAP}}(\hat{\mathbf{Y}}_{\text{anc}}))$ where $\hat{\mathbf{Y}}_{\text{anc}} \sim \mathcal{N}(\boldsymbol{\mu}_{\text{anc}}, \boldsymbol{\Sigma}_{\text{anc}})$. We will show that, when we set

$$\boldsymbol{\mu}_{\text{anc}} = \boldsymbol{\mu}_{\hat{\mathbf{Y}}}, \quad \boldsymbol{\Sigma}_{\text{anc}} = \boldsymbol{\Sigma}_{\hat{\mathbf{Y}}} + \boldsymbol{\Sigma}_{\hat{\mathbf{Y}}} \boldsymbol{\Sigma}_{\text{like}}^{-1} \boldsymbol{\Sigma}_{\hat{\mathbf{Y}}}$$

the distribution $P(\hat{Y}_{\text{MAP}}(\hat{\mathbf{Y}}_{\text{anc}}))$ becomes equal to the posterior distribution $\mathcal{N}(\boldsymbol{\mu}_{\text{post}}, \boldsymbol{\Sigma}_{\text{post}})$. There are three steps needed to show equality between these distributions:

1. Show that $P(\hat{Y}_{\text{MAP}}(\hat{\mathbf{Y}}_{\text{anc}}))$ is normally distributed for some mean and variance, $\boldsymbol{\mu}_{\text{post}}^{\text{RMS}}, \boldsymbol{\Sigma}_{\text{post}}^{\text{RMS}}$.

2. Show that $\boldsymbol{\mu}_{\text{post}}^{\text{RMS}} = \boldsymbol{\mu}_{\text{post}}$.

3. Show that $\boldsymbol{\Sigma}_{\text{post}}^{\text{RMS}} = \boldsymbol{\Sigma}_{\text{post}}$.

Using the same reasoning as above, in the limit as $M \to \infty$,

$$P(\hat{Y}_{\text{MAP}}(\hat{\mathbf{Y}}_{\text{anc}})) = \arg\max_{\hat{\mathbf{Y}}} P(\mathbf{y}_{\text{obs}}|\hat{\mathbf{Y}}) P_{\text{anc}}(\hat{\mathbf{Y}})$$

$$= \arg\max_{\hat{\mathbf{Y}}} \mathcal{N}(\hat{\mathbf{Y}}|\boldsymbol{\mu}_{\text{like}}, \boldsymbol{\Sigma}_{\text{like}}) \mathcal{N}(\hat{\mathbf{Y}}|\boldsymbol{\mu}_{\text{anc}}, \boldsymbol{\Sigma}_{\text{anc}})$$

Since the max of a Gaussian is the mean, then

$$F_{\text{MAP}}(\hat{\mathbf{Y}}_{\text{anc}}) = A\hat{\mathbf{Y}}_{\text{anc}} + b \tag{19}$$

where we define:

$$A = \boldsymbol{\Sigma}_{\text{post}} \boldsymbol{\Sigma}_{\hat{\mathbf{Y}}}^{-1} \tag{20}$$

$$b = \boldsymbol{\Sigma}_{\text{post}} \boldsymbol{\Sigma}_{\text{like}}^{-1} \boldsymbol{\mu}_{\text{like}} \tag{21}$$

We will now show that $\mathbb{E}[\hat{Y}_{\text{MAP}}(\hat{\mathbf{Y}}_{\text{anc}})] = \boldsymbol{\mu}_{\text{post}}$. Because we set $\boldsymbol{\mu}_{\text{anc}} = \boldsymbol{\mu}_{\hat{\mathbf{Y}}}$:

$$\mathbb{E}[\hat{Y}_{\text{MAP}}(\hat{\mathbf{Y}}_{\text{anc}})] = \mathbb{E}[A\hat{\mathbf{Y}}_{\text{anc}} + b]$$

$$= A\mathbb{E}[\hat{\mathbf{Y}}_{\text{anc}}] + b$$

$$= A\boldsymbol{\mu}_{\hat{\mathbf{Y}}} + b$$

$$= \boldsymbol{\Sigma}_{\text{post}} \boldsymbol{\Sigma}_{\hat{\mathbf{Y}}}^{-1} \boldsymbol{\mu}_{\hat{\mathbf{Y}}} + \boldsymbol{\Sigma}_{\text{post}} \boldsymbol{\Sigma}_{\text{like}}^{-1} \boldsymbol{\mu}_{\text{like}}$$

$$= \boldsymbol{\mu}_{\text{post}}$$

Finally, we will show that $\mathbb{V}\text{ar}[\hat{Y}_{\text{MAP}}(\hat{\mathbf{Y}}_{\text{anc}})] = \boldsymbol{\Sigma}_{\text{post}}$.

$$\mathbb{V}\text{ar}[\hat{Y}_{\text{MAP}}(\hat{\mathbf{Y}}_{\text{anc}})] = \mathbb{V}\text{ar}[A\hat{\mathbf{Y}}_{\text{anc}} + b]$$

$$= A\mathbb{V}\text{ar}[\hat{\mathbf{Y}}_{\text{anc}}]A^T$$

$$= (\boldsymbol{\Sigma}_{\text{post}} \boldsymbol{\Sigma}_{\hat{\mathbf{Y}}}^{-1})(\boldsymbol{\Sigma}_{\hat{\mathbf{Y}}} + \boldsymbol{\Sigma}_{\hat{\mathbf{Y}}} \boldsymbol{\Sigma}_{\text{like}}^{-1} \boldsymbol{\Sigma}_{\hat{\mathbf{Y}}})(\boldsymbol{\Sigma}_{\text{post}} \boldsymbol{\Sigma}_{\hat{\mathbf{Y}}}^{-1})^T$$

$$= (\boldsymbol{\Sigma}_{\text{post}} + \boldsymbol{\Sigma}_{\text{post}} \boldsymbol{\Sigma}_{\text{like}}^{-1} \boldsymbol{\Sigma}_{\hat{\mathbf{Y}}})(\boldsymbol{\Sigma}_{\hat{\mathbf{Y}}}^{-1} \boldsymbol{\Sigma}_{\text{post}})$$

$$= \boldsymbol{\Sigma}_{\text{post}} \boldsymbol{\Sigma}_{\hat{\mathbf{Y}}}^{-1} \boldsymbol{\Sigma}_{\text{post}} + \boldsymbol{\Sigma}_{\text{post}} \boldsymbol{\Sigma}_{\text{like}}^{-1} \boldsymbol{\Sigma}_{\text{post}}$$

$$= \boldsymbol{\Sigma}_{\text{post}} (\boldsymbol{\Sigma}_{\hat{\mathbf{Y}}}^{-1} + \boldsymbol{\Sigma}_{\text{like}}^{-1}) \boldsymbol{\Sigma}_{\text{post}}$$

$$= \boldsymbol{\Sigma}_{\text{post}}$$

So, $P(\hat{Y}_{\text{MAP}}(\hat{\mathbf{Y}}_{\text{anc}})) \sim \mathcal{N}(\boldsymbol{\mu}_{\text{post}}^{\text{RMS}}, \boldsymbol{\Sigma}_{\text{post}}^{\text{RMS}})$ where $\boldsymbol{\mu}_{\text{post}}^{\text{RMS}} = \boldsymbol{\mu}_{\text{post}}$ and $\boldsymbol{\Sigma}_{\text{post}}^{\text{RMS}} = \boldsymbol{\Sigma}_{\text{post}}$.

Thus, in the limit as $M \to \infty$, $P(\hat{Y}_{\text{MAP}}(\hat{\mathbf{Y}}_{\text{anc}})) = P(\hat{\mathbf{Y}}|\mathbf{y}_{\text{obs}})$.

$\square$

### A.3 DERIVATION OF LOSS FUNCTION

We start with

$$\boldsymbol{\phi} = \arg\max_{\boldsymbol{\phi}} \mathbb{E}_{\boldsymbol{\theta}_{\text{anc}} \sim P_{\text{prior}}(\boldsymbol{\theta}_{\text{anc}})} \log P_{\text{like}}(\mathbf{y}_{\text{obs}} \mid g(\mathbf{x}_{\text{sample}}, \boldsymbol{\theta}_{\text{anc}}; \boldsymbol{\phi})) + \log P_{\text{anc}}(g(\mathbf{x}_{\text{sample}}, \boldsymbol{\theta}_{\text{anc}}; \boldsymbol{\phi}))$$

If we choose $\boldsymbol{\mu}_{\text{anc}} = \boldsymbol{\mu}_{\hat{\mathbf{Y}}}$ and $\boldsymbol{\Sigma}_{\text{anc}} = \boldsymbol{\Sigma}_{\hat{\mathbf{Y}}}$, we can simplify the logarithm of the likelihood and anchor distributions as follows:

$$\log P_{\text{like}}(\mathbf{y}_{\text{obs}} \mid g(\mathbf{x}_{\text{sample}}, \boldsymbol{\theta}_{\text{anc}}; \boldsymbol{\phi})) = \sum_i \log \mathcal{N}(y_{\text{obs}}^i | g(x_{\text{sample}}^i, \boldsymbol{\theta}_{\text{anc}}; \boldsymbol{\phi}), \sigma_\epsilon)$$

$$= -\frac{1}{2\sigma_\epsilon^2} \sum_i \|y_{\text{obs}}^i - g(x_{\text{sample}}^i, \boldsymbol{\theta}_{\text{anc}}; \boldsymbol{\phi})\|_2^2$$

$$\log P_{\text{anc}}(g(\mathbf{x}_{\text{sample}}, \boldsymbol{\theta}_{\text{anc}}; \boldsymbol{\phi})) = \log \mathcal{N}(g(\mathbf{x}_{\text{sample}}, \boldsymbol{\theta}_{\text{anc}}; \boldsymbol{\phi})|\boldsymbol{\mu}_{\hat{\mathbf{Y}}}, \boldsymbol{\Sigma}_{\hat{\mathbf{Y}}})$$

$$= -\frac{1}{2}\boldsymbol{\delta}^T \boldsymbol{\Sigma}_{\hat{\mathbf{Y}}}^{-1} \boldsymbol{\delta}$$

where $\delta_j = g(x_{\text{sample}}^j, \boldsymbol{\theta}_{\text{anc}}; \boldsymbol{\phi}) - f(x_{\text{sample}}^j; \boldsymbol{\theta}_{\text{anc}})$.

Putting these together, we get:

$$\boldsymbol{\phi} = \arg\max_{\boldsymbol{\phi}} \mathbb{E}_{\boldsymbol{\theta}_{\text{anc}} \sim P_{\text{prior}}(\boldsymbol{\theta}_{\text{anc}})} -\frac{1}{2\sigma_\epsilon^2} \sum_i \|y_{\text{obs}}^i - g(x_{\text{sample}}^i, \boldsymbol{\theta}_{\text{anc}}; \boldsymbol{\phi})\|_2^2 - \frac{1}{2}\boldsymbol{\delta}^T \boldsymbol{\Sigma}_{\hat{\mathbf{Y}}}^{-1} \boldsymbol{\delta}$$

$$= \arg\min_{\boldsymbol{\phi}} \mathbb{E}_{\boldsymbol{\theta}_{\text{anc}} \sim P_{\text{prior}}(\boldsymbol{\theta}_{\text{anc}})} \frac{1}{N} \sum_i \|y_{\text{obs}}^i - g(x_{\text{sample}}^i, \boldsymbol{\theta}_{\text{anc}}; \boldsymbol{\phi})\|_2^2 + \frac{1}{N}\sigma_\epsilon^2 \boldsymbol{\delta}^T \boldsymbol{\Sigma}_{\hat{\mathbf{Y}}}^{-1} \boldsymbol{\delta}$$

