# OpenReview forum: "Generative Posterior Networks for Approximately Bayesian Epistemic Uncertainty Estimation"
_ICLR.cc/2022/Conference — ICLR 2022 Submitted_

### Official Review · Reviewer_15Ce · 2021-11-01

**Correctness:** 3
**Technical Novelty And Significance:** 3
**Empirical Novelty And Significance:** 2
**Recommendation:** 5
**Confidence:** 4

**Main Review:**

Strengths:

(1)	The proposed method proposes a novel idea to build a mapping function from prior to posterior. It avoids ensemble training and achieves reasonable uncertainty quantification results.

(2)	The proposed method addresses the high-dimensional parameter space issue by constructing the mapping function in low-dimensional output space.

Weakness:

For the methods: the theory part is not well-written. Some equations are confusing and may have possible mistakes.

(1) The derivation of Eq. 13 and Eq. 14 is confusing. Why the second term $\log P_{anc}(g(x_{sample,\phi,\theta_{anc}}))$ in Eq. 13 includes the posterior sample $g(x_{sample,\phi,\theta_{anc}})$ in the prior distribution? When looking into the details in Appendix.A.3, there might be some mistakes. For example, why $\log P_{anc}(g(x_{sample},\phi,\theta_{anc}))$ and $\log P_{like}(y_{obs}|g(x_{sample},\phi,\theta_{anc}))$ both equals to $\sum_{i} \log N(y_{obs}^i | g(x_{sample}^i,\phi,\theta_{anc}),\sigma_{\epsilon})$

(2) The proposed method does not clearly explain how to obtain the parameters of the prior distribution $P(\hat{Y})$, i.e.,  $\mu_{\hat{Y}},\Sigma_{\hat{Y}}$.

(3) When we use low-dimensional $z$ for sampling in Eq. 15, there is no explicit equation for $L_{reg}(z_1,z_2,...,z_k)$. The details of such regularization should be provided and there is no theoretical justification of why we need to add $L_{reg}(z_1,z_2,...,z_k)$ to keep $z_1,z_2,...,z_k$  roughly normally distributed.

(4) No training details are provided.

For the experiments:

(1) Could you explain why the ensemble 10 methods are significantly worse than the proposed method as the proposed method is supposed to only improve efficiency of the Bayesian ensemble method?  It also does not make sense that the MC dropout method outperforms the ensemble method.  Are there any theoretical justifications for such dramatical improvement in accuracy compared to anchor-regularized neural networks ensembles (Pearce et al., 2020)?

(2) For the image classification tasks, the paper evaluates the proposed method based on the out-of-distribution prediction accuracy and the confidence interval. The confidence interval for classification tasks is confusing. The authors should provide more analyses of why those metrics can be used for evaluating uncertainties.

(3) The author should provide some analyses for the convergence of the model. The model may be hard to train especially when the mapping function $g$ is not powerful enough. Additional ablation studies should also be provided such as the sensitivity of hyperparameters.

(4) No experiments for comparing the efficiency and complexity of different methods. From the introduction section, it seems that a major contribution of the paper is to improve efficiency.





**Summary Of The Paper:**

The paper develops an uncertainty quantification method as an extension of the Bayesian ensemble method. Instead of training multiple ensemble models through MAP optimization, the proposed method tries to learn a mapping function between the prior distribution and the posterior distribution of model parameters, which avoids the complex training of ensemble models and achieves better efficiency. Due to the high-dimensional parameter space, the authors propose to learn the mapping function in the low-dimensional output space with theoretical justifications. The experiments include image classification and Bayesian optimization, where authors compare the proposed method with Bayesian ensemble and MC-dropout in terms of uncertainty estimation.

**Summary Of The Review:**

Overall, this paper proposes interesting ideas by building a mapping function from the prior to the posterior. However, the authors did not explain their ideas well and there might be some mistakes in their formulation. The author may also need to provide more detailed empirical results and corresponding analyses.

---

> ### Author Response · Authors · 2021-11-17
> **Response to criticisms/suggestions**
>
> Thank you so much for your thoughtful review. We are still working on addressing some of the comments, but we wanted to respond before the end of the discussion period to hopefully get some back and forth.
>
> Regarding your question about equation 13, the generative model is taking the place of \theta in equation 8. In this equation, we want to replace each MAP solution with a generative model that maps directly from anchor points to MAP solutions. It turns out there was a typo in equation 9 that may have caused this confusion. Equation 9 should have been as follows:
>
> $\phi = \text{argmax}\_{\phi} E\_{\theta\_{\text{anc}} \sim P\_{\text{anc}}(\theta\_{\text{anc}})} \log P\_{\text{like}}(y\_{\text{obs}} \mid g(\theta\_{\text{anc}}; \phi)) + \log P\_{\text{anc}} (g(\theta\_{\text{anc}}; \phi))$
>
>
> Great catch on the typo in the proof. We’ve fixed that now.
>
> To address your point about the lack of explanation of the embedding regularization, we’ve added some details to the main text. The reason why it’s important to regularize the embeddings towards a normal distribution is simply to make it easier to sample from (otherwise we wouldn’t know the distribution of the embedding space).
>
> A common criticism among these reviews is the lack of experiment details. We will make sure to add additional details to make the experiments clearer and easier to replicate.
>
> Regarding your point about the poor performance of Ensemble-10, we found an issue with our code that resulted in poor uncertainty prediction for Ensemble-10. We will update the tables and ROC curves for the next version (we’re still running those experiments again).
>
> Regarding your question about the confidence interval for classification, the main idea with this measurement is simply to illustrate how well the different methods quantify uncertainty. We would expect that, when the model is very uncertain, the posterior distribution should have non-negligible probability mass on every class. Conversely, when the model is very certain, the posterior distribution should have non-negligible probability mass **only** on a single class. This is the notion we hope to capture with the CI-All and CI-Any measures. We will make this motivation clearer.
>
> You, along with the other reviewers, mentioned it would be helpful to add a scalability experiment. We will add a new figure to the paper showing out-of-distribution prediction ROC-AUC vs training time for both our method and the ensembles.
>
> Regarding your point about adding a convergence experiment, we agree. We plan to add a plot of the out-of-distribution prediction performance during training for different hyper-parameters.

---

> > ### Comment · Reviewer_15Ce · 2021-11-21
> > **Responces to the Authors**
> >
> > Thanks for the authors’ responses. We noticed that the authors have corrected some mistakes in their theory parts. However, the rebuttal is mainly for illustration purposes and clarification of misunderstandings. The significant changes in theories need another round of review. Moreover, the experiment results are still not empirically sound. The scalability experiments are missing, and the poor Ensemble-10 results have not been addressed. Hence, we think the current version of this paper is not ready for publication and I will give it a score of 4.

---

> > > ### Author Response · Authors · 2021-11-23
> > > **Final Changes**
> > >
> > > Sorry for the confusion. We actually hadn't updated the draft when you sent this message. Our original response was sent as we were still working on the new draft.
> > >
> > > We've now updated the draft of the paper. Unfortunately, we weren't able to address all of your suggestions in time, namely the scalability experiment and additional experiment details. We have added a small-scale experiment, a visualization of sampling from the embedding space, fixed the ROC curves for Ensemble-10, and tried to clarify some important information brought up by you and other reviewers. We hope the newest draft is clearer and we will make sure to add the scalability experiment and additional experiment details to the final version.

---

### Official Review · Reviewer_v6ij · 2021-11-01

**Correctness:** 3
**Technical Novelty And Significance:** 2
**Empirical Novelty And Significance:** 3
**Recommendation:** 5
**Confidence:** 4

**Main Review:**

# Strengths

This paper provides a nifty alternative to ensembling, which significantly reduces the computational complexity as only one network needs to be trained now. Moreover, as opposed to trying to get the posterior over parameter space the authors compute the posterior over function space; this drastically reduces the number of dimensions the GPN needs to predict, making the problem much easier to learn.

# Weaknesses

While I am a fan of the approach conceptually, I think there are a number of weaknesses.

Firstly, I think the related works section is lacking. I think most of the discussion on MCMC and VI could have been cut out. More time should have been focusing on other prior works on ensembling neural networks.
As the approach can be seen as a form of hypernetwork[https://arxiv.org/abs/1609.09106], I think this should have definitely been discussed by the authors. There is also other work--similar to Pearce et al--showing how to approximately sample from the posterior of neural networks using the NTK which is also pertinent to discuss[https://arxiv.org/abs/2007.05864].

Next, I am confused by section 4.1 as a whole. To be specific and ensure that I am on the same page as the authors, I am going to rewrite the problem formulation (i apologize in advance for the lack of boldface. for some reason it isn't working properly for me).

We have training data,
$$ (x_{obs}, y_{obs}) = \{ (x_{obs}^i, y_{obs}^i) \}_{i=1}^N $$

where the following relationship is assumed
$$ y_{obs}^i = f(x_{obs}^i; \theta) + \varepsilon_i, \quad \epsilon_i \sim \mathcal{N}(0, \sigma_\epsilon). $$

In supervised Bayesian learning, the goal is to compute the following posterior where I will first write the posterior over the parameters

$$ p(\theta \vert x_{obs}, y_{obs} )  \propto P_{prior}(\theta) \prod_{i=1}^N P_{like} (y_{obs}^i \vert x_{obs}^i, \theta)  $$

One can also compute a posterior over functions as well.
Let
$$ \tilde{y}^i_{obs} = f(x_{obs}^i; \theta) $$
thus we can express the relationship between the inputs and outputs as
$$ y_{obs}^i = \tilde{y}^i_{obs} + \epsilon_i. $$

Let $ \tilde{Y} = (\tilde{y}^1_{obs}, ..., \tilde{y}^N_{obs}) $, we now want to compute the following posterior
$$ p(\tilde{Y} \vert x_{obs}, y_{obs}) \propto P_{prior}(\tilde{Y} \vert x_{obs}) \prod_{i=1}^N P_{like}(y^i_{obs} \vert \tilde{y}^i_{obs}) $$
In Lemma 1, the authors prove that in function space, the likelihood is normally distributed with respect to $\tilde{Y}$. This is trivially true by construction  i.e., $P_{like}(y_{obs}^i \vert \tilde{y}^i) = \mathcal{N}(y_{obs}^i \vert \tilde{y}^i, \sigma_\epsilon)  $. When operating in parameter space, the likelihood is complicated but the prior is easier; when moving to function space, this switches.

Besides that, Lemma 1 also makes no sense to me. In the lemma, they take a discretization of the input space but I see no reason to do that as the likelihood is only defined on the inputs provided in the training data. This is also true for theorem 1 as well. In my opinion, section 4.1 could have been entirely removed, which would have left more space for more experiments.

Concerning the experiments section, it was disappointing to not see an experiment similar to Pearce et al, comparing the true posterior to the posterior obtained by GPN on a simple toy dataset. As many assumptions were made, I think it is very important that the approach was empirically verified in a setting where the ground truth can be obtained. Next, I don't understand experiment 2. Regardless of the learning method used, the posterior studied in this paper revolves around using supervised data. I have no idea how the authors were able to use the unlabeled data nor does it seem fair to me.

**Summary Of The Paper:**

In this work, the authors introduce generative posterior networks (GPN). GPN is a single network that approximately produces samples from posterior over neural network parameters; this is in contrast to the standard approach of training N individual neural networks.

**Summary Of The Review:**

While I like the idea, I think more work needs to be done on the structure of the paper and the experiments. I think section 4.2 could be cut, allowing for more space to do experiments and a more fleshed-out discussion section.

---

> ### Author Response · Authors · 2021-11-17
> **Response to criticisms/suggestions**
>
> Thank you so much for the effort you put into this review. We are still working on addressing some of the comments and running additional experiments for a new draft of the paper, but we wanted to respond before the end of the discussion period to hopefully get some back and forth.
>
> We think many of your criticisms come from a misunderstanding of the difference between $x_{\text{sample}}$ and $x_{\text{obs}}$. Since this is a crucial part of our method, we should have made this distinction clearer. $x_{\text{obs}}$ is the vector of inputs from the observed data and $x_{\text{sample}}$ are uniformly sampled points from the input distribution used for constructing the transformed random vector $\hat{Y}$. This is important because we are trying to measure epistemic uncertainty so we are particularly focused on the posterior at the points not in the observed data. We’ve updated the paper to emphasize this distinction. With this distinction, do Lemma 1 and Theorem 1 make sense and is their importance clear?
>
> We agree that adding in a low-dimensional ground truth experiment would illustrate our method’s accuracy and also help the readers' understanding of our method. We plan to add this experiment in the new version of the paper.
>
> Thank you for the notes on the related work. We will try to incorporate those suggestions in the next version of the paper. The one part we disagree with is the hyper-network discussion. Although our method is inspired by a hyper-network, we reformulate the problem as a standard generative model in Section 4.1.
>
> A common criticism among these reviews is the lack of experiment details. We will make sure to add additional details to make the experiments clearer and easier to replicate.

---

> > ### Comment · Reviewer_v6ij · 2021-11-17
> > **Response to reviewer**
> >
> > Thank you so much for replying and I'm excited to see the coming updates.
> >
> > Concerning section 4.1, I'm now sadly more confused so I want to spend time to try and understand what exactly you're trying to show and to hopefully converge onto a clearer section. Before I go into any math I first want to make sure I'm on the same page. My understanding after your reply is that the goal of section 4.1 is that to show that the posterior predictive distribution retrieved using RMS is equivalent to the true posterior predictive distribution. Am I on the same page?

---

> > > ### Author Response · Authors · 2021-11-17
> > > **Response to reviewer**
> > >
> > > Thank you for taking the time to try to understand this!
> > >
> > > Yes, the goal of 4.1 is to both introduce the new formulation of the Bayesian Inference problem, estimating the posterior of the outputs of $f$ on some sample points rather than the parameters of $f$, and then to show that the posterior distribution retrieved using RMS is equivalent to the true posterior predictive distribution in this new Bayesian Inference problem.

---

> > > > ### Comment · Reviewer_v6ij · 2021-11-17
> > > > **Response to reviewer**
> > > >
> > > > Thanks, now I'm on the same page!
> > > > I don't think you need all of this extra machinery to demonstrate what you want to show and I also think Lemma 1 is wrong.
> > > >
> > > > For simplicity, let's assume the same conditions that are used in Pearce et al., i.e. we are in the Bayesian linear regression setting and that we are given training, $(x_{obs}, y_{obs})$ and query points, $x_{sample}$.
> > > > We can write the posterior predictive distribution using Bayes' rule as
> > > > $$ p(\hat{Y} \vert x_{obs}, y_{obs}, x_{sample}) \propto p(y_{obs} \vert \hat{Y}, x_{obs}, x_{sample}) p(\hat{Y} \vert x_{sample}) $$
> > > > As was demonstrated in section 2 of *Gaussian Processes in Machine Learning*, Bayesian linear regression is equivalent to  Gaussian process regression using a linear kernel, i.e. $k(x, x') = x^\top x'$.
> > > > So we can compute exactly the prior and likelihood in the above posterior.
> > > > The prior is
> > > > $$p(\hat{Y} \vert x_{sample}) = \mathcal{N}(0, K_{s, s})$$
> > > > where $K_{s, s}$ is the kernel matrix computed over $x_{sample}$.
> > > > Using equation 2.19 in *Gaussian Processes in Machine Learning*, we can write the likelihood as
> > > > $$ p(y_{obs} \vert \hat{Y}, x_{obs}, x_{sample}) = \mathcal{N}(K_{o, s}K_{s, s}^{-1} \hat{Y}, K_{o, o} - K_{o, s}K_{s, s}^{-1}K_{s, o} + \sigma^2 I )$$
> > > > where $K_{o, s}$ and $K_{o, o}$ is definied simirlarly as $K_{s, s}$ and $I$ is the identity matrix.
> > > > Because the prior is Gaussian and the likelihood is linear in $\hat{Y}$, then you can very easily use the same math done in Pearce et al to show that RMS works in this case.
> > > > Note that my use of easily doesn't mean I don't think this is cool. I think this was very nifty on the authors' part to think of this!
> > > >
> > > > Also, I can't see how Lemma 1 is true. $p(y_{obs} \vert x_{obs}, \sigma^2) = \mathcal{N} ( x_{obs}^\top \theta, \sigma^2 I)$ where the covariance matrix is purely diagonal while the covariance of $p(y_{obs} \vert \hat{Y}, x_{obs}, x_{sample})$ does have correlations!

---

> > > > > ### Author Response · Authors · 2021-11-23
> > > > > **Response to reviewer**
> > > > >
> > > > > If I understand your comment correctly, you are saying that if we are assuming our function $f$ is linear, then Theorem 1 becomes much easier to prove. We agree, but our theorem doesn't require $f$ to be linear. Right after equation (6) we make a note that this assumption is not needed for our proof. We do, however, need to assume that the outputs are approximately Gaussian when the parameters are normally distributed (which will be exact when the penultimate layer is infinitely wide). We are not aware of a standard result saying that Bayesian Inference on the output of a neural network is equivalent to regression on some kernel.
> > > > >
> > > > > Regarding your counter example for Lemma 1, once the x_sample becomes dense enough, the observations should become conditionally independent given $\hat{Y}$. As a simple example of this, consider the case where $x_{\text{obs}} = x_{\text{sample}}$. In this case,
> > > > > $ P(y\_{\text{obs}} | \hat{Y}, x\_{\text{obs}}, x\_{\text{sample}}) = P(y_{\text{obs}} | f(x\_{\text{obs}} | \theta), x\_{\text{obs}}, x\_{\text{sample}}) = \mathcal{N}(y_{\text{obs}} | f(x\_{\text{obs}} | \theta), I \sigma\_{\epsilon}) = P(y\_{\text{obs}} | \theta, x\_{\text{obs}}) $
> > > > > The goal of Lemma 1 is to show that in the general case (where $x_{\text{obs}} \ne x_{\text{sample}}$) as some of the elements of $ x\_{\text{sample}}$ get closer and closer to elements in $ x\_{\text{obs}} $, $ P(y\_{\text{obs}} | \hat{Y}, x\_{\text{obs}}, x\_{\text{sample}})$ gets closer to $ P(y\_{\text{obs}} | \theta, x\_{\text{obs}})$.

---

> > > > > > ### Comment · Reviewer_v6ij · 2021-11-29
> > > > > > **Response**
> > > > > >
> > > > > > I should have been more principled in my math. To be more precise, if the last layer is infinitely wide then the output of the DNN is a Gaussian process that holds for randomly initialized, untrained networks [1] and for trained networks [2, 3]. As is discussed in chapter 2 of Gaussian Processes in Machine Learning, GP regression is equivalent to linear regression where the basis function corresponding to the kernel is used. For DNNs, this basis function was found in [3]. Thus, the math that I wrote in my previous comment does hold for DNNs in the NTK regime, which is the same operating regime that this work is assuming in their proof; similar mathematics was used in [4]. I apologize, I should have been more precise in what I meant by the assumption that f is linear in $\theta$.
> > > > > >
> > > > > > I also don't quite understand the response concerning lemma 1. As I pointed out in my response, the functional forms of the two distributions are different. Moreover, the correlations in the Grammian matrix depend on the inputs, not the outputs.
> > > > > >
> > > > > >
> > > > > > I appreciate the changes that have been made to the manuscript. I think the introduction of the first figure helps a lot and I am happy to see a regression example added. I am somewhat disappointed to not see an updated related works section. I also don't see any details on architectures or training details. I will raise my score but I do think this paper would benefit from more time spent on it.
> > > > > >
> > > > > >
> > > > > > [1] Radford M. Neal. Bayesian Learning for Neural Networks. PhD thesis, University of Toronto, Dept. of Computer Science, 1994b.
> > > > > > [2] Jacot et al. Neural Tangent Kernel: Convergence and Generalization in Neural Networks. NeurIPS 2018.
> > > > > > [3] Lee et al. Wide Neural Networks of Any Depth Evolve as Linear Models Under Gradient Descent. NeurIPS 2019.
> > > > > > [4] He at al. Bayesian Deep Ensembles via the Neural Tangent Kernel. NeurIPS 202.

---

> ### Author Response · Authors · 2021-11-23
> **Final Changes**
>
> We've now updated the draft of the paper. Unfortunately, we weren't able to address all of your suggestions in time, namely the scalability experiment. We have added a small-scale experiment, a visualization of sampling from the embedding space, and tried to clarify some important information brought up by you and other reviewers. We hope the newest draft is clearer and we will make sure to add that experiment in the final version.

---

### Official Review · Reviewer_vyt5 · 2021-11-05

**Correctness:** 2
**Technical Novelty And Significance:** 2
**Empirical Novelty And Significance:** 2
**Recommendation:** 3
**Confidence:** 2

**Main Review:**

Pros:
- The paper deals with an important area of research that provides an efficient way to estimate epistemic uncertainty.
- The method provides somewhat improved results compared to previous methods.

Cons:
- Even though this work is built upon Pearce et al 2020, it largely hinges on the previous work in the manuscript. Specifically, it is hard to understand without knowing several definitions and results in Pearce. So, I’d recommend authors to provide self-contained backgrounds and results in ch.3. Also, the assumed relationship that parameter-likelihood is equal to data-likelihood (Eq 3) seems to be a special case, but there is no comments on that.
- It is unclear why the authors exclude the input variables X in the data/parameter-likelihoods.
- Theoretical results and mathematical states in the paper seem not rigorous. Specifically, in the statements and proof of lemma 1 and theorem 1, the authors simply assume equality between two probability distributions in the limit. This can have many potential problems, such as limit/integral interchangability. It would be better to write and develop their theoretical results in terms of convergence behavior.
- I have also concerns regarding experiments. First of all, the experiment section lacks important experimental details that make reproducing the results in the paper hard. Also, while the primary motivation for this research is to remove the MAP optimization procedure, there is no comparison to baselines in terms of computational complexity. The last paragraph in page 7 is not reasonable since having many unlabeled samples in practice cannot be the reason why the authors have access to test samples during training. In addition, the CI-ALL measure (that requires the same answers to all members of ensemble) seems unnatural since the power of ensemble comes from diversified predictions from individual members and mostly their majority voting results are of interest. Finally, I cannot understand how the authors could compute the classification accuracy for OOD, unknown class samples given fixed architecture.


**Summary Of The Paper:**

This paper built upon Pearce et al 2020 that shows regularizing parameters to prior distributions in ensemble of NNs results in the approximate Bayesian inference, aiming to improve computational efficiency of the previous method. Specifically, in the method in Pearce et al, sampling one parameter from posterior requires to find a MAP solution for some anchor point, which can be computationally demanding. Therefore, the authors propose to learn MAP function from an anchor point to MAP solution, which removes the required MAP optimization procedure.


**Summary Of The Review:**

The paper is not self-contained. Also, even though their motivation for the improvement is to improve the computational efficiency, the experiments did not provide any results on that. There are some issues in the mathematical statements and proof thereof.

---

> ### Author Response · Authors · 2021-11-17
> **Response to criticisms/suggestions**
>
> Thank you for your review. We really appreciate the effort you put into this. We are still working on addressing some of the comments and running additional experiments for a new draft of the paper, but we wanted to respond before the end of the discussion period to hopefully get some back and forth.
>
> There seems to have been some confusion around the definitions of data-likelihood vs parameter-likelihood. These functions are defined to have the same output for the same $\theta$ and $y_{\text{obs}}$, but the data-likelihood takes $y_{\text{obs}}$ as an input and $\theta$ as a parameter while the parameter-likelihood takes $\theta$ as an input and $y_{\text{obs}}$ as a parameter. This distinction is necessary since the product of two Gaussian functions is also a Gaussian if they are defined on the same variable.
>
> Regarding your question about dropping the $y_{\text{obs}}$ term from the likelihood, we chose to drop $y_{\text{obs}}$ to make the equations more legible. We will make this explanation explicit in the next version of the paper.
>
> You, along with the other reviewers, mentioned it would be helpful to add a scalability experiment. We will add a new figure to the paper showing out-of-distribution prediction ROC-AUC vs training time for both our method and the ensembles.
>
> That’s a great point about the proof. It should be quite easy to show uniform convergence in Lemma 1 and thus, make the limit/integral interchangability more rigorous. We’re still working on this and plan to have it added by the end of the discussion period.
>
> A common criticism among these reviews is the lack of experiment details. We will make sure to add additional details to make the experiments clearer and easier to replicate.
>
> Regarding your point about the use of unlabeled samples, there seems to have been some confusion. We are not using test samples during training. During training we provide labeled data containing in-distribution data and unlabeled data containing both out-of-distribution and in-distribution data (i.e. data from the entire training set). The test data we use (both in-distribution and out-of-distribution) is never seen during training. We tried to make this clearer in our new draft.
>
> We disagree on the lack of validity of the CI-All measure. We agree that ensembles are often used to improve accuracy of predictions, but this is not the primary goal of ensembles in the context of epistemic uncertainty estimation. In this context, the discrepancy between the members of the ensemble is used to measure uncertainty. Thus, we want this discrepancy to be low when the model is confident. We use CI-All for in-distribution test data as a way of measuring how confident the model is on in-distribution data.
>
> Regarding OOD prediction accuracy, we should have made this clearer. In fact, it is not a measure we really care about, so we’ve decided to remove it from paper for clarity. However, to clarify, we construct a network with 10 outputs (1 for each class), but only provide labeled data for 6/10 of those classes during training. In our next version, we plan to remove the additional 4 outputs from the network (which will require us to redefine the CI-Any measure) so as not to give the model any additional information.

---

> ### Author Response · Authors · 2021-11-23
> **Final Changes**
>
> We've now updated the draft of the paper. Unfortunately, we weren't able to address all of your suggestions, namely the scalability experiment and proving uniform convergence for Lemma 1. We have added a small-scale experiment, a visualization of sampling from the embedding space, and tried to clarify some important information brought up by you and other reviewers. We hope the newest draft is clearer and we will make sure to add those experiments in the final version.

---

### Official Review · Reviewer_ZgaQ · 2021-11-06

**Correctness:** 4
**Technical Novelty And Significance:** 3
**Empirical Novelty And Significance:** 3
**Recommendation:** 6
**Confidence:** 3

**Details Of Ethics Concerns:**

No potential ethics concerns.

**Main Review:**

This paper introduces a new proposal to make use of ensemble NN methods, GPNs. With the approximations imposed, GPNs perform well in the experiments shown. Although some points about the text could be improved and there are certainly some potentially important drawbacks, it seems an interesting contribution to the field. There seems to be a couple of tests that need to be done, but the results point in the right direction.

# Pros:

* Allow to obtain samples in a more scalable fashion from NN-ensemble-like models.
* The results obtained support the proposed method in a solid manner when compared to previous methods.

# Cons:

* The conditions imposed on the method makes its predictions behave in a Gaussian-like manner. This could be fine in cases where data does not exhibit complex behaviour, but could be a problem for cases with multimodality, heteroscedasticity, etc.
* Notation can be improved to make the description of the method clearer.


# Other questions / comments:

* It could be argued that, contrary to what is pointed out in the introduction, VI or GP-based methods can be easy to work with. In fact, if some assumptions are feasible or the data behaves in a certain manner, they can be pretty illustrative and informative models. I would suggest rewriting this part of the introduction, pointing out more clearly why NN ensembles may be more interesting in some cases.

* GPs are a model class, not an inference method.

* I would suggest conducting an experiment on the scalability of the method, measuring convergence times with big datasets. This would help us gauge correctly how well does the method perform in this matter as well.

* While it is true that basic BNNs degrade their performance in higher dimensionality settings, there have been important advances in the last years to prevent this from happening, especially when using deeper or wider networks. This can be said for the function-space-based optimization systems, such as those of [1,2].

* As a mere suggestion, I find the notation of the data-likelihood and parameter-likelihood PDF unnecessarily complex. I would either simplify this to improve the clarity of the text or explain the notation further (not just relying on referenced sources). In general, through the paper, the notation could be clearer.

* Figure 3 can be improved. I suggest trying alternative visualizations s.a. combining different boxplots into one using colours, or at the very least increasing the font (in the CIFAR-10 case) to better show the results.

* Have the authors conducted any tests in regression datasets? It could be very interesting to see the resulting behaviour in those cases.

 ## References

 [1] Ma, C., Li, Y., and Hernández-Lobato, J. M. (2019). “Variational implicit processes”. In: International Conference on Machine Learning, pp. 4222–4233.

 [2] Sun, S., Zhang, G., Shi, J., and Grosse, R. (2019). “Functional variational Bayesian neural networks”. In: International Conference on Learning Representations.


**Summary Of The Paper:**

NN ensembles are useful models to estimate epistemic uncertainty in the predictions. Since their computational cost is generally high, they introduce Generative Posterior Networks (GPNs) as a generative NN model to approximate the posterior distribution of the inference process. Under not very restrictive assumptions, GPNs work as well as previous methods, if not better. This is shown through extensive experiments, especially in classification problems.

**Summary Of The Review:**

The contribution is novel and significative, although there are a couple of things that should be addressed in the revised version of the paper.

---

> ### Author Response · Authors · 2021-11-17
> **Response to criticisms/suggestions**
>
> Thank you so much for your thoughtful review. We are still working on addressing some of the comments and running additional experiments for a new draft of the paper, but we wanted to respond before the end of the discussion period to hopefully get some back and forth.
>
> You, along with the other reviewers, mentioned it would be helpful to add a scalability experiment. We will add a new figure to the paper showing out-of-distribution prediction ROC-AUC vs training time for both our method and the ensembles.
>
> Regarding your suggestion about a regression experiment, we completely agree. We plan to add a regression task in the next version.
>
> Thank you for your suggestions regarding the notation. We’re working on clearing up some of the notation that caused confusion for you and other reviewers.
>
> And thank you for your suggestions regarding the related work section and the 3. We will try to incorporate those changes for the next version.

---

> ### Author Response · Authors · 2021-11-23
> **Final Changes**
>
> We've now updated the draft of the paper. Unfortunately, we weren't able to address all of your suggestions, namely the scalability experiment and the regression experiment. We have added a small-scale experiment, a visualization of sampling from the embedding space, and tried to clarify some important information brought up by you and other reviewers. We hope the newest draft is clearer and we will make sure to add those experiments in the final version.

---

### Decision · Program_Chairs · 2022-01-20

**Decision:**

Reject

**Comment:**

This article proposes a novel uncertainty quantification method, formulating the problem as a Bayesian inference problem. Instead of training multiple ensemble models through MAP optimisation, as in ensemble methods, the proposed approach tries to learn a mapping function between the prior distribution and the posterior distribution of model parameters. This avoids the complex training of ensemble models and achieves better efficiency.

The approach is novel, and the problem of importance. The paper however suffers from a number of weaknesses:
* Some theoretical results would need to be made mathematically more rigorous
* The presentation is unclear and confusing in some places
* Empirical results are not reproducible due to the lack of details
Although the authors clarified some of the points raised by reviewers in their response, the paper in its current form is not ready for publication, and I recommend rejection.